# Progressive Exaptation of Endogenous Retroviruses in Placental Evolution in Cattle

**DOI:** 10.3390/biom13121680

**Published:** 2023-11-21

**Authors:** Toshihiro Sakurai, Kazuya Kusama, Kazuhiko Imakawa

**Affiliations:** 1School of Pharmaceutical Science, Ohu University, 31-1 Misumido, Koriyama 963-8611, Fukushima, Japan; 2Department of Endocrine Pharmacology, Tokyo University of Pharmacy and Life Sciences, 1432-1 Horinouchi, Hachioji 192-0392, Tokyo, Japan; kusamak@toyaku.ac.jp; 3Research Institute of Agriculture, Tokai University, 9-1-1 Toroku, Higashi-Ku, Kumamoto 862-8652, Japan; ik459102@tsc.u-tokai.ac.jp

**Keywords:** cattle, placenta morphology, ERVs (endogenous retroviruses)

## Abstract

Viviparity is made possible by the placenta, a structure acquired relatively recently in the evolutionary history of eutherian mammals. Compared to oviparity, it increases the survival rate of the fetus, owing to the eutherian placenta. Questions such as “How was the placenta acquired?” and “Why is there diversity in placental morphology among mammalian species?” remain largely unsolved. Our present understanding of the molecules regulating placental development remains unclear, owing in no small part to the persistent obscurity surrounding the molecular mechanisms underlying placental acquisition. Numerous genes associated with the development of eutherian placental morphology likely evolved to function at the fetal–maternal interface in conjunction with those participating in embryogenesis. Therefore, identifying these genes, how they were acquired, and how they came to be expressed specifically at the fetal–maternal interface will shed light on some crucial molecular mechanisms underlying placental evolution. Exhaustive studies support the hypothesis that endogenous retroviruses (ERVs) could be evolutional driving forces for trophoblast cell fusion and placental structure in mammalian placentas including those of the bovine species. This review focuses on bovine ERVs (BERVs) and their expression and function in the placenta.

## 1. Introduction

Animals reproduce their offspring in three different ways: oviparous, in which the eggs hatch from externally laid eggs; ovoviviparous, in which the eggs are retained in the parent’s body and hatch just before birth; and viviparous, in which the eggs are kept in the body of the parent (usually the mother) without a shell and are born alive. Viviparity, which enables the maternal maintenance of the embryonic environment, is a reproductive strategy observed in various groups of vertebrates, including bony fishes, elasmobranchs, amphibians, reptiles, and mammals [1]. This primitive structure and function of the placenta, facilitating viviparity, was acquired relatively early in the course of vertebrate evolution, with a substantial fraction of vertebrate species developing this capability [2,3].

How did mammals evolve from egg-laying to in utero incubation, and how did the proto-placental function (primitive viviparous) develop into the mature placenta of today? The transition from ovoviviparity to viviparity and the subsequent emergence of placentation must have required considerable changes in the morphology and physiology of the reproductive tract. For example, monotremes, including the platypus and echidna, are oviparous mammals. They lay thin, low-mineral eggs that typically hatch about ten days after being laid. During a six-month period in the uterus, the developing young is nourished by maternal secretions through a simple yolk sac placenta [4]. Therian mammals, a lineage that encompasses marsupials and eutherians (all extant mammals except monotremes and marsupials), evolved the capacity for live birth (viviparity) during this evolutionary process. They abandoned both the mineralized eggshell and yolk and ultimately developed a placenta. It is worth noting that marsupials significantly differ from eutherians in that their gestation period is relatively short (an average of 25 days), with pregnancy essentially completed within a single estrous cycle. In contrast, eutherian pregnancies are prolonged, lasting up to 670 days (with an average of 131 days) without estrous cycles [4]. The oldest eutherian, *Juramaia sinensis*, acquired a placenta approximately 160 million years ago [5], while the oldest known mammal to date, *Adelobasileus cromptoni*, has been dated to about 225 million years ago [6]. In the 65 million years between the birth of the first mammals and the acquisition of the placenta, did they acquire new genes to form the placenta? If so, how did mammals acquire those genes? Or did mammals express and exapt existing genes in the placenta? It was thought that placentation occurred only once in the common ancestor of mammals, but it is now thought to have happened independently many times in already-diverged classes and families of mammals [7]. Thus, the acquisition of the placenta in vertebrates represents clear cases of convergent evolution, both at the macro level such as in organ acquisition, and at the molecular level such as through gene acquisition and exaptation [8].

Many of the genes that characterize placental morphology in eutherians likely evolved to be expressed at the fetal–maternal boundary, concurrently with embryogenesis. Consequently, identifying these genes and the mechanisms through which they acquired specific expression at the fetal–maternal interface should illuminate essential molecular mechanisms underlying placental evolution. Moreover, conducting a comprehensive comparative study of genes expressed in the placenta across various animal species may unveil common genes necessary for primitive placental formation or placentation, as well as sets of genes that have either been newly acquired or have lost their functions in different species. Mika and colleagues compared the genes used by the human endometrium with those expressed by the endometria of 32 other species, including monkeys, marsupials, other mammals, birds, and reptiles [9]. The analysis revealed that humans use nearly 1000 genes that other animals do not use [9]. These genes are involved in placental invasion, vascular growth, and immune system regulation [9]. Elucidating the genetic and molecular mechanisms underlying these phenotypic differences between species will lead to a better understanding of the diversity of placental morphology.

## 2. Diversity and Classification of Placenta in Mammals

Although the role of the placenta is to nurture the fetus, no other mammalian organ has a structure as different between lineages as the placenta. The genes responsible for viviparity could serve as the foundation for the initial development of primitive placenta during mammalian evolution, with subsequent independent acquisitions of new genes in various species leading to species-specific placental morphologies (Figure 1). Eutherian placentas can be classified based on several criteria, including their shape (number and distribution of nutrient exchange regions on the placental surface), invasiveness (number of maternal tissue layers separating maternal blood and fetal tissue), interdigitation (degree of contact between fetal and maternal tissue in nutrient exchange regions), placental weight relative to neonatal weight, and the relative direction of maternal and fetal blood flow [10]. In fact, eutherian placentas can be histologically classified according to the degree of syncytialization (mode of distribution of villi) into epitheliochorial placenta (diffuse placenta) in pigs and horses, synepitheliochorial placenta (cotyledonary placenta) in cattle and sheep, endotheliochorial placenta (zonary placenta) in dogs and cats, and hemochorial placenta (discoidal placenta) in humans and mice (Table 1). This classification depends on the morphology of the fetal–maternal interface. Syncytialization is the fusion of trophoblast cells to form syncytia (syncytiotrophoblasts) on the surface of the chorioallantoic membrane, as in the human placenta. The most important barriers to elucidating the origin of evolutionarily novel and placental phenotypes are the absence of transient structures among extant lineages and a lack of experimental systems that allow for detailed functional studies. Differences in gene expression underlie changes in anatomical structure, suggesting that changes in gene expression at the maternal–fetal interface underlie pregnancy traits specific to each species. On the other hand, genes commonly expressed in connection with placentas of all mammals may be the genes responsible for proto-placentation (Figure 1). In fact, Armstrong et al. identified a set of 115 core genes expressed in the placenta of 14 mammalian species [11].

## 3. The Principle of Genomic Rewiring during Waves of Retroviral Infection

There have been several major genome-size expansions in the lineage from the beginning of life to the rise of human beings. Two primary mechanisms responsible for increasing genome size are recognized: DNA duplication and the incorporation of retrotransposons. DNA duplication events have led to major innovations in the history of life, giving rise to eukaryotes (2 billion years ago), multicellular organisms (1 billion years ago), and vertebrates (500 million years ago). Subsequently, there was an explosion of retrotransposons about 50 to 40 million years ago, which is believed to have driven species diversification and increased the complexity of reproductive strategies, including the diversification of placental morphology [15]. Traditionally, retrotransposons have received little attention in discussions of life’s evolution. However, it has become increasingly evident that retrotransposons play a significant role in shaping genomes, leading to outcomes such as genome size expansion and various effects on genetic function. These effects encompass the disruption of gene function through insertion and the potential for genes to acquire new functions. Thus, genomes have co-evolved with transposable elements (TEs) and have devised strategies to regulate uncontrollable elements while extracting novel functions from their newly inserted nucleotides. 

Although different types of TEs are found in various organisms, all major TEs can be classified into two main categories: elements that can be transposed via a DNA intermediate and a cut-and-paste mechanism (transposons), and those using an RNA and a copy–paste mechanism (retrotransposons) [16]. Given that TEs make up a substantial portion of animal genomes, comprising approximately 27% in the bovine genome [17], they are instrumental in driving evolutionary changes in genome size and composition [18]. Host organisms share numerous repetitive sequences, encompassing genes, cis-regulatory elements, and chromatin domain boundaries. These shared sequences have the potential to alter gene regulatory networks and, moreover, are partially responsible for morphological evolution as in mammals [19]. Endogenous retroviruses (ERVs) belong to the TE and are considered to have originated from long terminal repeat (LTR)-type retrotransposons [20]. The genetic changes underlying the evolution of gestation appear to be associated with TEs that invaded and became integrated into early mammalian genomes. ERVs, in particular, are thought to have originated from ancient viruses that infected primordial germ cells or their products, including oocytes, sperm, and fertilized eggs in mammals and other vertebrates [21]. ERVs have become integral components of their host genome, resulting from waves of retroviral infection in the host’s ancestors, insertion of retroviral sequences into germline DNA, and vertical transmission from one generation to the next [21]. Remarkably, retrovirus-derived sequences account for 8.8% of the human genome [22], 10% of the mouse genome [23], and 18% of the bovine genome [24]. A study on the location and expression of bovine ERVs, as well as the characteristics of neighboring genes, has revealed that up to 1610 bovine genes contain bovine ERVs (BERVs) inserted within their introns [25]. Most of these BERVs are oriented in the antisense direction, and several genes located in proximity to BERVs appear to be associated with viral response and chromatin assembly [25].

ERVs are genomic elements present in a wide range of vertebrates, spanning from basal vertebrates like sharks and rays to mammals [26]. Although LTR-type retrotransposons share close sequence similarities with ERVs, the envelope (*env*) genes retained by ERVs enabling transmitted particles to infect other cells and to replicate successfully. Many ERVs become inactivated through nucleotide insertions, deletions, substitutions, and epigenetic modifications, including DNA methylation and histone modifications [27,28]. Nevertheless, some ERV open reading frames (ORFs) are expressed as virus-derived proteins within host cells [29]. Over time, hosts have undergone alterations in these sequences, limiting their ability to reintegrate randomly into the genome (transposition) and producing virus-like particles with reduced effects or proteins with functions distinct from those of the host. As Haig [12] points out, the phenotypic effects of a gene in one organism can influence the fate of the same gene in another organism, highlighting the significance of genes’ evolutionary roles over the organisms themselves. The appearance of changes in the expression of existing genes is the result of changes in evolutionary interests across species and can be predicted as a functional evolution of the placenta, but this alone cannot explain the full diversity observed in placental morphology. Furthermore, it is possible that maternal–fetal conflicts have led to the removal of once-infected ERVs. Additionally, newly acquired genes, including species-specific ERVs, may further contribute to the generation of morphological diversity in the placenta (as illustrated in Figure 1).

## 4. *Peg10*, an LTR-Type Retrotransposon Commonly Expressed in Marsupial and Eutherian Placentas

The placenta is highly diverse among mammalian species, yet little is known about how it has evolved and what consequences the evolution of specific placental structures has for the mother and fetus. The acquisition of placental-like structures is thought to be due to the acquisition of the *paternally expressed gene 10* (*Peg10*) derived from an LTR-type retrotransposon about 160 million years ago [30]. Knockout mice lacking the *Peg10* show early embryonic lethality with placental defects, indicating the importance of this gene in placental development [31]. This gene is highly conserved among mammalian species and is evolutionarily new, being present only in marsupials such as kangaroos and eutherian groups such as humans and mice. In addition, an essential paralog of *Peg10*, *Peg11/retrotransposon-like 1* (*Rtl1*), is conserved only in eutherians [32], and this gene is essential for maintaining fetal capillaries [33]. *Peg10* and *Peg11*/*Rtl1* are imprinted genes believed to have originated from a Ty3/Gypsy LTR-type retrotransposon family, which encodes *gag*- and *pol*-like domains [30,31,32,33]. *Peg10* is found in both marsupials and eutherian lineages and contributes to trophoblast cell growth during early placentation [11]. On the other hand, *Peg11/Rtl1* is exclusive to eutherians and plays a critical role in maintaining endothelial cells in fetal capillaries during late embryonic development. It is evident that both eutherian *Peg10* and *Peg11/Rtl1* are key factors in establishing viviparity. However, it is apparent that these two genes alone place few constraints on the diversity of placental morphology.

## 5. The *env* Genes, Syncytin and Syncytin-like, Essential for Placentation in Mammals

Many studies have shown that ERVs play an important role in developing eutherian placentas and trophoblast cells and suggest the possibility that different species may have utilized ERVs of different origins or of the same origin during evolution. It is hypothesized that by utilizing newly acquired ERVs, each species has evolved the placenta independently (Figure 1), although maternal–fetal conflict through the use of TEs other than ERVs and selective pressures on existing and newly acquired genes cannot be excluded. Independently acquired *syncytin* genes have become integrated into the genome of humans [34,35,36,37,38], mice [26,39,40], rabbits [41], dogs [42], cats [42], sheep [43,44,45,46,47], cattle [13,48,49,50,51,52], or marsupials [53], which have been acquired independently in mammalian orders or species. All identified *syncytin* genes in different orders of mammals are unrelated, although they may share similar functional characteristics. However, both the ERV evolutionary pathway and the extent to which ERVs function in placentation remain unknown.

## 6. Morphology of the Bovine Placenta

In contrast to primates such as humans and monkeys, as well as mice, in most ruminants, implantation into the endometrial epithelium and subsequent placentation do not occur immediately after blastocyst formation [54]. The fertilized egg spends a long period in the uterine lumen before finally attaching to the endometrial epithelium and prompting the subsequent formation of placental structures. Ruminants such as cows and sheep have a unique placental variation known as synepitheliochorial placentas, that develop pre-determined sites of the uterine wall termed caruncles and are characterized by the presence of about 100 placentomes, a unique structure with both embryonic–maternal tissues [55]. These placentas can be considered to have a semi-invasive type of placentation [55]. The ‘syn-’ prefix signifies the contribution of the trophoblast binucleate cell (BNC)-derived syncytial trophoblast, while the term ‘epitheliochorial’ underscores the presence of extensive regions with simple placental cell attachment in the definitive placenta [56]. Bovidae is divided into seven subfamilies, including the subfamily Bovine, the subfamily Goatinae, and the subfamily Impala, and the placentas of all subfamilies are plexiform polyplacental, consisting of numerous placental segments. Histologically, the fetal side of the placenta has three characteristic cell types in ruminants: uninucleate trophoblast cells (UNCs), BNCs, and trinucleate cells (TNCs) [57]. In cattle and other ruminants, BNCs begin to emerge around day 20 of gestation. They originate from UNCs through processes such as mitotic polyploidization or endoreduplication [55,58]. This process continues throughout gestation, primarily in cotyledons after day 30, resulting in a consistent 15 to 20% of the trophectoderm being comprised of BNCs at various gestational stages [59]. In the placentome, which consists of maternal caruncles and fetal cotyledons, BNCs are believed to migrate while maintaining tight junctions with other trophoblast cells. Subsequently, they fuse with a uterine luminal epithelium (LE) cell, leading to the formation of TNCs in the sub-epithelial-stromal areas [55,58]. This allows secretory granules containing unique placenta-specific hormones, such as placental lactogen and pregnancy-associated glycoproteins, as well as microvesicles, including exosomes, and other factors, to be released via exocytosis into the endometrial stroma throughout gestation [55,60]. Among the bovine family, the form of cell fusion has been considered to differ among animal species, and only the Bovinae subfamily, to which bovines and water buffalo belong, and the Goatinae subfamily, to which goats and sheep belong, exhibit a unique characteristic in which cells on the fetal and maternal sides fuse together. In the bovine subfamily, BNCs and endometrial cells fuse one-to-one to form TNCs, while in the goat subfamily, multiple BNCs fuse with one endometrial cell to form multinucleate cells [58]. This characteristic of cell fusion in the placenta of bovine subfamily animals is thought to be due to bovine endogenous retrovirus K1 (BERVK1, also known as Fematrin-1), described below [51]. It has long been believed that the BNCs fuse with endometrial cells on the surface of the placenta on the maternal side, which invade the endometria and transport hormones to the mother, thereby maintaining pregnancy [58]. It was recently shown by two laboratories that the fusion between trophoblast cells in sheep is not limited to the formation of BNCs, and new BNCs involve the lateral fusion between growing syncytial plaques [61,62].

## 7. Expression of *env*-Derived Genes from Endogenous Retroviruses in Bovine Placenta

Although the placenta performs the same function in all mammals, the *env*-derived genes responsible for trophoblast fusion vary among mammalian species and had remained largely unidentified in bovines.

*Syncytin-Rum1* [50] is an *env*-gene, a part of ERV, expressed in BNCs which presumably entered the genome by infecting a cell in the germline, of which an envelope was required to perform this function. This integration occurred more than 30 million years ago in the common ancestor of ruminants [50]. *Syncytin-Rum1* has undergone a purifying selection, remaining conserved in most ruminants, except for Tragulidae (mouse-deer), whose ruminant ancestor diverged from others about 50 million years ago [50] (Table 2). While *Syncytin-Rum1* is expressed in both bovine and sheep placentas, its expression level is notably higher in sheep compared to cattle [50]. BERV-K1 *env* is also expressed in binucleated trophoblasts during cattle gestation and, 25.3 to 18.3 million years ago, BERV-K1 infected only the bovine subfamily ancestors [51]. It is probable that the structural similarity seen among ruminants’ placentas is due to the function of Syncytin-Rum1. The integration of both *Syncytin-Rum1* and *BERV-K1* into the bovine genome (as shown in Figure 1 and Table 2) suggests the possibility of successive integrations of ERVs with homologous functions, at least in the bovine. Furthermore, while *Syncytin-Rum1* exhibits high expression in the colon, cecum, rectum, kidney, mammary gland, and testis, *BERV-K1* is expressed in most tissues (Figure 2). These findings imply that newly acquired ERVs could contribute more effectively to placental morphogenesis or cellular function than pre-existing genes. Consequently, we refer to these consecutive ERV acquisitions as the ‘baton pass’ hypothesis. For more detailed information, refer to the following review [14].

While Syncytin-Rum1 exhibits fusogenic activity specifically under acidic conditions [51], BERV-K1 exhibits a much higher cell-fusion activity than Syncytin-Rum1 under physiological conditions [52,63]. It is possible that BERV-K1 became a key player in bovine placentation (Table 2). Interestingly, BERV-K1 also played a role in the formation of trophoblast hybrid cells in heterologous trophoblast cells (sheep trophoblast cells) [62]. In addition to *Syncytin-Rum1* and *BERV-K1*, other ERV *envs* such as *BERV-K2* [49], *bERVE-A* [50], and *bERVE-B* [50] have been identified. BERV-K1 and BERV-K2 share similar amino acid sequences and belong to the Beta retrovirus genus [65]. *BERV-K2* encompasses all coding sequences, including *gag*, *pro*-*pol*, and *env*, whereas *BERV-K1* retains only the *env* coding sequence [49]. *BERV-K1* and *BERV-K2* are expressed throughout different stages of pre-implantation development [66]. Their higher expression levels are detected in embryonic blastomeres (from the 2-cell to 16-cell stages), with a significantly lower ERV expression observed at the more differentiated blastocyst stage [66]. Notably, BERV-K2 *env* does not induce cell-to-cell fusion with endometrial cells [52,63], attributed to a failure of *env* glycoprotein maturation [63]. These varying properties of ERV *envs* may contribute to differences in cell fusion between animal species.

Since the release of the bovine genome information, numerous in silico analyses have been conducted to detect ERVs [13,25,49,51,67,68,69]. A search for intact *env* genes in the *Bos taurus* genome (*Bos taurus*_UMD3.1 version) identified 18 candidates from 5 endogenous retrovirus families, including *Syncytin-Rum1* [51] and another study identified *BERV-K1* [49] and *BERV-K2* [49]. Three computational tools (BLAST, LTR_STRUC, and Retrotector©) were used for the genome-wide detection of endogenous retroviruses (ERVs) in the bovine genome (Bos taurus_Btau_3.1) [67]. A total of 13,622 putative ERVs were initially identified, but only 1532 were further identified using two or three additional programs [67]. Our group similarly analyzed the bovine genome (Bos taurus_Btau 4.0) and identified 7624 ERV-derived genes, including 1542 *env*-derived genes [13]. Of these BERV *envs*, 18.4% (284 genes) were expressed in conceptus on day 22 of pregnancy, and approximately 4% (63 genes) of the *env*-derived genes in the genome were detected on all days studied (days 17, 20, and 22 of pregnancy) [13]. Of these *env*-derived genes, the sequence of the *env*-derived gene with the longest ORF (designated BERV-P *env*) was similar to the *Syncytin-Car1* gene found in dogs and cats [13]. It is thought that *BERV-P* integration occurred about 16.9 to 7 million years ago [13]. However, BERV-P *env* was not expressed specifically in the peri-implantation conceptus but in almost all tissues examined, like *BERV-K1* (Figure 2). These results suggest that placentation depends on a variety of retrovirus-derived genes that may have replaced endogenous predecessor genes during evolution and could answer the question of placental diversity in ruminants [14]. 

ERVs comprise three different classes depending on their relationship with their exogenous counterparts [70]: Class I is related to *Epsilonretrovirus* and *Gammaretrovirus*; Class II to *Alpharetrovirus*, *Betaretrovirus*, *Deltaretrovirus*, and *Lentivirus*; and Class III to *Spumavirus*. In cattle, Xiao et al. had identified sequences from different retrovirus families (*BERVγ4*, *γ7*, *γ9*—class I, and *BERVβ3*—class II) using a specialized PCR technique targeting the *pro*/*pol* complex [68]. These retrovirus families were named based on their resemblance to sheep ERVs [68]. *BERVγ4* was the most common family, originating approximately 13.3 million years ago and found to be distantly related to gamma retroviruses [71]. While full-length *BERVγ4* provirus sequences were found in specific chromosomes, partial sequences were distributed throughout the bovine genome [71]. The bovine genome also contained a full-length *BERVβ1* provirus with a standard organization [72]. The most conserved *BERVβ3* provirus contained multiple stop codons and closely resembled the *HERV-K* human family [73]. Whether these ERVs are expressed in the bovine placenta has not been evaluated. Re-examination of the bovine genome (*Bos taurus*_UMD3.1 version) using LTRharvest and LTRdigest further detected class I *BoERV25*-*BoERV27* and class II *BoERV28*-*BoERV30* [69]. However, whether these ERVs are expressed in the bovine placenta has not been examined. The availability of new bovine genomic data on ERVs will help to confirm or discover new insights into the evolutionary implications of ERVs demonstrated in model animals.

*Endogenous Jaagsiekte retroviruses* (*enJSRV*) exist in the bovine genome [48], but whether *enJSRV* is expressed in the bovine placenta has not been investigated. It has been reported that enJSRV *env* promotes trophoblast cell fusion in the sheep placenta through the activation of the PKA/MEK/ERK1/2 signaling pathway [74].

bERVE-A *envs* have been identified as transcripts in the bovine placenta and trophectoderm [50]. *bERVE-A* exhibited approximately 94% homology to a sequence similar to human *syncytin* and featured an ORF comprising 107 amino acids [50]. bERVE-A *env* was highly expressed in post-implantation conceptuses (D20 and D22), with no expression observed in tissues or organs other than the ovary, testis, rectum, or spleen (Figure 2). Expression of *bERVE-A* was detected at a low level in the conceptuses from day 17 to 19 of gestation, with an increase in expression observed in fetal membranes up to day 30 of gestation [50]. *bERVE-A* is expressed preferentially in BNCs [57]. In addition, *bERVE-A* was rarely expressed in the endometrium throughout the estrus cycle [50]. This gene contains a *syncytin 1*-like SU domain and ASCT2 binding domain, but it does not possess fusogenic activity due to the loss of the fusion peptide [50]. It is suggested that bERVE-A may have arisen as a result of BNC formation rather than serving as an initiator of cell-to-cell fusion, as it lacks an intact envelope sequence [50]. On the other hand, *bERVE-B* is expressed in most organs, but its expression in conceptuses was very weak (Figure 2). Differences in these retroviral gene integrations and degrees of expression may account for some changes in the placental structures and/or functions among the placentas of different ruminants. 

## 8. Regulatory Mechanisms Involved in BERV Gene Expression in the Bovine Placenta

In the human and mouse placentas, Suppressyn (SUPYN), originated from a retrovirus distinct from *Syncytin-1*, interacts with the syncytin-1 receptor (ASCT2/SLC1A5), leading to the inhibition of cytotrophoblast cell fusion [75]. It should be noted, however, that neither *SUPYN* nor any *SUPYN*-like sequences are present in the bovine genome. Consequently, the relationship between SUPYN and BERVs could not be discussed. In a study by Kitao et al. [76,77], functional syncytin post-transcriptional regulatory elements (SPREs) were identified in several *syncytin* genes. These elements were suggested to enhance the expression of viral genes previously repressed due to inefficient codon frequencies or repressive elements within the coding sequence. This discovery raises the possibility that SPRE sequences are present in *syncytin*-like genes, including *BERV-K1*, *Syncytin-rum1*, and *BERV-P* in bovines. Additionally, while a transcription factor binding site for retinoid X receptor alpha (Rxra), a steroid and thyroid hormone receptor, has been examined in the thyroid gland, its presence or function in the uterus or placenta remain unexplored [25]. This underscores the need for further investigation into the regulatory mechanisms governing ERV gene expression in these specific contexts.

The expression of human ERV (HERV) is intricately regulated at the LTR level, where these LTRs act as promoters for HERV expression [78]. They contain essential RNA polymerase II regulatory sequences [79,80] and numerous transcription factor binding sites [81], through which LTRs can interact with nuclear transcription factors [82]. Notably, recombination events have led to the presence of solo LTRs in the human genome, excising the rest of the provirus [83]. Remarkably, as much as 85% of HERVs have been removed through these recombination events [84], and most HERV loci now exist as solo LTRs. These solo LTRs have the ability to serve as promoters in both sense and antisense directions [85], potentially influencing host gene expression [86,87]. As a result, LTRs play a crucial role in epigenetic modifications, ultimately governing the regulation of both HERV and human gene expression. In the human placenta, DNA methylation levels are generally reduced compared to other tissues, reflecting a higher proportion of HERV LTRs functioning as tissue-specific promoters within placental tissue [88]. In particular, the CpG island of the 5’ LTR is hypomethylated in placental cells, contrasting with its hypermethylated state in other tissues [89]. It should be noted that the investigation into ERV silencing and transcriptional regulation has primarily focused on humans. It is essential to acknowledge that while the mechanisms and principles governing ERV transcriptional regulation in one species may provide insights into another, there can be significant differences among species. Therefore, studies on the regulation of BERV expression by LTRs must be examined using the bovine placenta and bovine cells.

As shown in Figure 2, ERV expression in cattle is not exclusive to the placenta. It is conceivable that ERVs may have played roles in various organs or cell types, with some later adapting to serve in placental functions. To understand their function in the placenta, it is important to identify the cell types expressing these ERVs in different organs and their respective functions. While in silico analysis has identified the expression of numerous ERVs in the early bovine placenta, the specific roles of individual ERVs in placentation remain unclear. Additionally, there has been limited research into the regulatory mechanisms governing ERV expression in cattle. One reason for this lack of clarity is the scarcity of studies on the cellular and molecular mechanisms of bovine placentation compared to other animal models. Elucidation of the cellular and molecular mechanisms governing trophoblast differentiation and function in the bovine placenta will undoubtedly contribute to a more detailed understanding of the role of ERVs. A recent study [90] demonstrated the mechanism associated with the differentiation of trophoblast MNCs into BNCs. Additionally, there have been reports of the successful establishment of bovine trophoblast stem cells capable of generating two distinct functional trophoblast cell types [91].

Moreover, our ongoing research focuses on generating trophectoderm cells from bovine induced pluripotent stem (iPS) cells and identifying expressed ERVs. The data we collect will significantly contribute to elucidating the ERVs responsible for placental morphology characteristics in bovine species and potentially beyond.

## 9. Expression of *gag*-Derived Genes from Endogenous Retroviruses in Cattle Placenta

ERVs consist of three primary genes: *gag*, encoding the capsid protein; *pro*-*pol*, encoding enzymes for maturation, replication, and insertion; and *env*, encoding the envelope protein [70]. *BERV-K1*, *BERV-K2*, *bERVE-A*, *Syncytin-Rum1*, and *BERV-P* have all been reported to originate from *env* regions homologous to syncytin. However, ERVs from other regions, such as *gag* and *pol*, which may also play a role in ruminant placentation, had not been identified or characterized. In a search for ERV-derived nucleotide structures expressed in the bovine peri-implantation conceptuses [13], we identified ten putative ERV *gags* located between functional genes in the bovine genome [14]. One candidate ERV gene with *gag*/*pol* on bovine chromosome 7 exhibited minimal expression on day 20 during uterine epithelium attachment, increased on day 22 at the onset of placentation, and maintained high expression until at least day 150 [14]. Its expression in bovine trophoblast cells was induced by WNT agonists, a common intracellular signal in placental genes [14]. We named this ERV *gag*/*pol* BERV-K3, which, like other BERVs, was expressed not only in the placenta but also in the skin, kidney, liver, and ileum [14]. Therefore, understanding the function of BERV-K3 in these organs may provide insights into the divergence and role in placental function. In addition to *BERV-K3*, we have identified several other ERVs containing the *gag*/*pol* region expressed during the peri-implantation period of bovine embryos [14]. Our current research involves the gene expression and cDNA cloning of these ERVs with *gag*/*pol* in peri-implantation conceptuses and various organs.

## 10. Conclusions

Throughout the course of evolution, organisms underwent a transition from asexual to sexual reproduction, and subsequently, from oviparity to viviparity. This transition required the emergence of the uterus as a vital organ for nurturing developing fetuses and physiological changes. During this pivotal period, it is conceivable that various retroviruses spread across mammalian species, integrating into their genomes. Prior to the emergence of the placenta, it is plausible that various retroviruses indeed spread across mammalian species and became integrated into their genomes. Over time, these proviruses likely underwent both negative and positive selection, eventually acquiring functional roles in various cell types and organs. It is important to note that most ERVs are not limited to specific tissues; instead, they exhibit varying degrees of expression across a wide range of tissues and organs. In essence, ERVs with pre-existing functions in diverse tissues or organs may have been co-opted to operate at the fetal–maternal boundary within the uterus throughout the course of pregnancy evolution. This adaptation enabled new functions, such as cell-fusion activities, to emerge gradually over time, ultimately contributing to the remarkable diversity we observe in placental structures today. Despite the challenges associated with elucidating the functions of ERV genes, particularly due to the absence of homologous genes in model organisms like mice, the shared viral ancestry of ERVs suggests the possibility of shared functional similarities at the molecular level. A comprehensive investigation of this hypothesis could shed light on the evolutionary history of the placenta and unravel the mysteries surrounding its diversity.

## Figures and Tables

**Figure 1 biomolecules-13-01680-f001:**
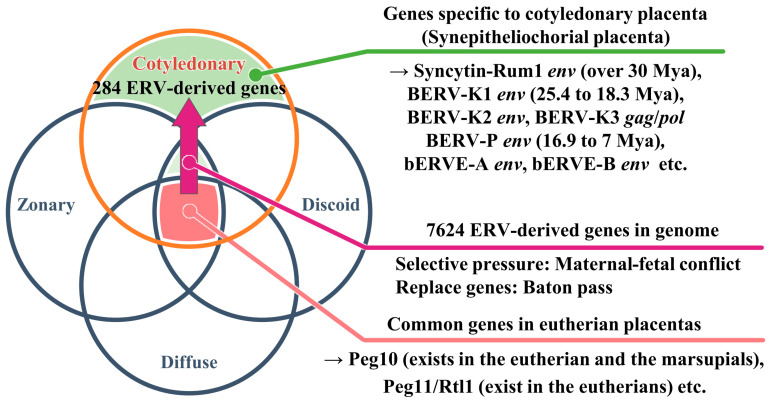
Hypothesis on the diversity of placental morphology by ERVs [11,12,13,14]. Mechanisms underlying placental innovations encompass the recycling of proteins already expressed within the tissue (co-option), the induction of gene expression typically observed elsewhere in the organism (recruitment), and the introduction of novel genes into the organism’s genome through gene duplication or retroviral insertion (horizontal transfer). It was hypothesized that mammals have developed species-specific placental morphology by harnessing the uniquely acquired ERVs within each lineage. Peg10 and Peg11/Rtl1, responsible for fundamental placental function, were present in common ancestral mammals. Subsequently, the bovine acquired the retrovirus-derived genes individually. We have identified 7624 ERV-derived genes in the bovine genome [12]. Among these, 284 ERVs, including *BERV-K1*, *BERV-K2*, *BERV-K3*, *Syncytin-Rum1*, *BERV-P*, *bERVE-A*, and *bERVE-B*, are expressed in the cotyledonary placenta due to epigenetic regulations, maternal–fetal conflict [13], and gene replacement through a Baton pass [14]. These factors collectively contribute to the development of ruminant-specific placental morphology. Mya: million years ago.

**Figure 2 biomolecules-13-01680-f002:**
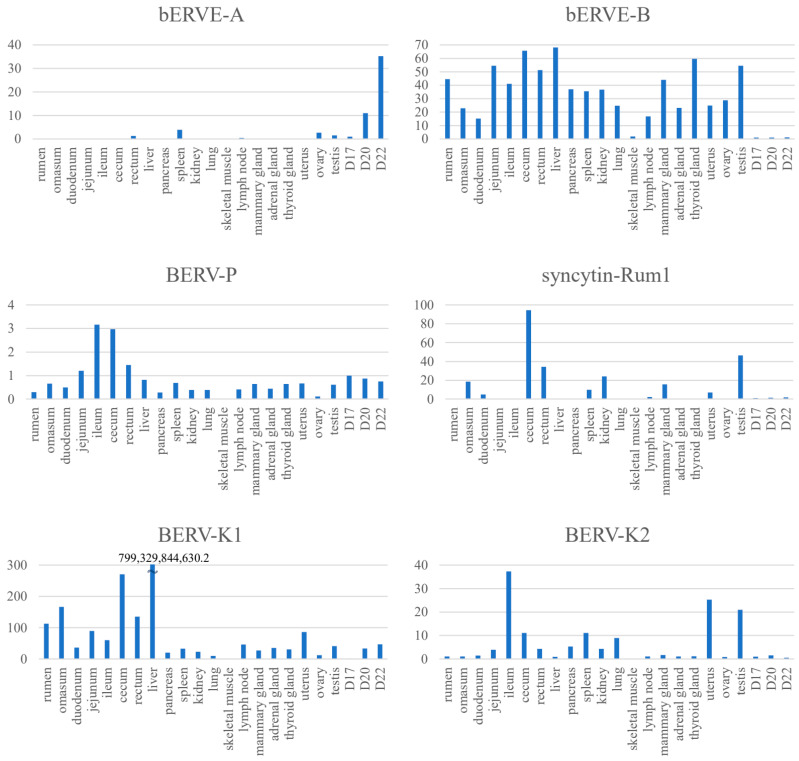
Relative expression of BERV *env* genes in bovine tissues and conceptuses. Total RNA was extracted from the tissues of Japanese black cattle (gestation day 190) at NIAS, Ibaraki, Japan, excluding the uterus, ovary, or testis. mRNA from the uterus, ovary, and testis was sourced from Zyagen, San Diego, CA, USA. Conceptuses (days 17, 20, and 22, with day 0 denoting the day of estrus) were collected at Zen-noh Embryo Transfer Center, Hokkaido, Japan (for more details, refer to [13]). For reverse transcription, 500 ng of total RNA was used, following the manufacturer’s manual of the ReverTra Ace qPCR RT Master Mix with gDNA Remover (Toyobo, Osaka, Japan). The resulting cDNA was diluted 1:10 with DNase/RNase-free water and subsequently employed for qPCR with specific primers. The primer sequences were designed based on previously published information [13,50,51]. qPCR was conducted using the THUNDERBIRD Next SYBR qPCR Mix (Toyobo) and the Thermal Cycler Dice Real Time System (Takara Bio, Shiga, Japan). Relative expression levels of target genes were determined using the Delta-Delta Ct method and normalized to GAPDH mRNA expression. The results were presented relative to day 17 of the conceptus, which was set as 1.

**Table 1 biomolecules-13-01680-t001:** Classification of mammalian placentas.

Mode of Distribution of Villi	Degree of Syncytialization	Representative Species	Type of Fused Cells
Cotyledonary	Synepitheliochorial	Cow, Sheep	Fetomaternal hybrid
Diffuse	Epitheliochorial	Horse, Pig	None
Zonary	Endotheliochorial	Dog, Cat	Syncytiotrophoblast
Discoid	Hemochorial	Mouse, Human	Syncytiotrophoblast

**Table 2 biomolecules-13-01680-t002:** ERV genes supposed to play roles in placentation in ruminants.

	Appearance in Host Genome	Species	Expressed Tissues (Localization) ^#1^	Function	References
Syncytin-Rum1 *env*	Over 30 Mya	Most ruminants(except for the family *Tragulidae*)	Placenta (BNCs)	Fusogenic activity	[51]
BERV-K1 *env*	25.4 to 18.3 Mya	Bovinae subfamily	Placenta (BNCs), embryonic blastomeres (two- to 16-cell stage)	Fusogenic activity	[52,63]
BERV-K2 *env*	No data	No data	Endometrium, embryonic blastomeres (2- to 16-cell stage)	No data	[49,63]
BERV-P *env*	16.9 to 7.0 Mya	genus Bos	Conceptus during periattachment (Trophectoderm)	No data	[13]
bERVE-A *env*	No data	*Bos taurus*	Placenta (BNCs)	No data	[50]
bERVE-B *env*	No data	No data	Endometrium, placenta	No data	[50]
BERV-K3 *gag*/*pol*	No data	No data	Conceptus after attachment (Trophectoderm), placenta	No data	[64]

#1: Other than showing in Figure 2. Mya: million years ago.

## Data Availability

Not applicable.

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
