# Peer review of "Progressive Exaptation of Endogenous Retroviruses in Placental Evolution in Cattle"

_biomolecules, 2023, doi:10.3390/biom13121680_

Round 1
Reviewer 1 Report (Previous Reviewer 4)
Comments and Suggestions for Authors
The authors have responded adequately to my suggestions and the manuscript is now, in my opinion, suitably adapted for Biomolecules.
Comments on the Quality of English LanguageMinor adjustment to English language is still recommended to improve the style, word usage, and fluency.
Author Response
We thank the reviewer for carefully reading our manuscript. Prior to the last submission, the second revision, this paper had been proofread by a native speaker of English, and a native speaker has also proofread this third version.
Reviewer 2 Report (New Reviewer)
Comments and Suggestions for Authors
This is a well-written, comprehensive review on a fascinating topic. It will be a valuable addition to the literature.
There are a few points at which the text is somewhat imprecise; these should be corrected. On line 171, it states that env genes enable particle formation. In fact retroviral particle formation is completely independent of env proteins; on the other hand, particles released from one cell will only be able to successfully infect another cell if they possess an envelope. Thus the envelope is essential for successful replication via transmission of particles between cells, but not for particle formation.
The statement on lines 215-220 is very confusing. It refers to syncytin as “a gene” but then says that the sequences of syncytins vary between species. It is my understanding that the syncytins in different orders of mammals are unrelated; while they may share functional characteristics, they are distinct “genes” which have been acquired independently in different orders.
The abbreviation “BNC” (line 232) is never defined.
Another confusing phrase is line 268: “Syncytin-Rum1 is env-derived ERV…” This mixes up the endogenous retrovirus, which presumably entered the genome by infecting a cell in the germline and required an envelope to perform this infection, with the envelope gene (or protein encoded by the gene) of the virus.
It is surprising to read (line 300) that the failure of BERV-K2 to induce fusion with endometrial cells is “attributed to a failure of env glycoprotein maturation (65)”. As far as I could see, ref. 65 says nothing about maturation, and merely documents the rather low fusogenic activity of this envelope.
Line 383 mentions a transcription factor binding site for “Rxra”. What is this? It requires some explanation.
Author Response
Comments: This is a well-written, comprehensive review on a fascinating topic. It will be a valuable addition to the literature.
Response: We thank the reviewer for thoroughly reading the paper and providing detailed, valuable comments. We carefully evaluated the comments and have revised the manuscript accordingly. Our responses are given in a point-by-point manner below. Changes are shown in the manuscript in red.
Comments: There are a few points at which the text is somewhat imprecise; these should be corrected. On line 171, it states that env genes enable particle formation. In fact retroviral particle formation is completely independent of env proteins; on the other hand, particles released from one cell will only be able to successfully infect another cell if they possess an envelope. Thus the envelope is essential for successful replication via transmission of particles between cells, but not for particle formation.
Response: We agree with this comment. To improve clarity, we have changed the sentence as follows: Although LTR-type retrotransposons share close sequence similarities with ERVs, the envelope (env) genes retained by ERVs enable transmitted particles to infect other cells and to replicate successfully. (revised manuscript, page 4, lines 170-172)
Comments: The statement on lines 215-220 is very confusing. It refers to syncytin as “a gene” but then says that the sequences of syncytins vary between species. It is my understanding that the syncytins in different orders of mammals are unrelated; while they may share functional characteristics, they are distinct “genes” which have been acquired independently in different orders.
Response: The reviewer is right to point out that syncytin genes have been acquired independently in different orders or species. Accordingly, we have revised this statement on lines 217-222 as follows: Independently acquired syncytin genes have become integrated into the genome of humans [32-36], mice [37-39], rabbits [40], dogs [41], cat [41], sheep [42-46], cattle [47-52] or marsupials [53], which have been acquired independently in mammalian orders or species. All identified syncytin genes in different orders of mammals are unrelated, although they may share similar functional characteristics. However, both the ERV evolutionary pathway and the extent to which ERVs function in placentation remain unknown.
Comment: The abbreviation “BNC” (line 232) is never defined.
Response: Based on the reviewer’s comment, we have defined it. The revised version is as follow: ‘the trophoblast binucleate cell (BNC)’. (Revised manuscript page 6, line 234)
Comments: Another confusing phrase is line 268: “Syncytin-Rum1 is env-derived ERV…” This mixes up the endogenous retrovirus, which presumably entered the genome by infecting a cell in the germline and required an envelope to perform this infection, with the envelope gene (or protein encoded by the gene) of the virus.
Response: We have changed the subtitle and the sentences as follows: Subtitle: Expression of env-genes derived from Endogenous Retroviruses in Bovine Placenta. Lines 270-273: Syncytin-Rum1 [50] is an env-gene, a part of ERV, expressed in BNCs which presumably entered the genome by infecting a cell in the germline, of which an envelope was required to perform this function. This integration occurred more than 30 million years ago in the common ancestor of ruminants [50].
Comments: It is surprising to read (line 300) that the failure of BERV-K2 to induce fusion with endometrial cells is “attributed to a failure of env glycoprotein maturation (65)”. As far as I could see, ref. 65 says nothing about maturation, and merely documents the rather low fusogenic activity of this envelope.
Response: The references were misnumbered. Reference [65] was corrected to [64]. (Revised manuscript page 7, line 303)
Comments: Line 383 mentions a transcription factor binding site for “Rxra”. What is this? It requires some explanation.
Response: Rxra is retinoid X receptor alpha. We have added the following explanation to page 9, lines 384-387 of the revised manuscript. ‘Additionally, while a transcription factor binding site for retinoid X receptor alpha (Rxra), a steroid and thyroid hormone receptor, has been examined in the thyroid gland, its presence or function in the uterus or placenta remains unexplored [22].’ (Revised manuscript page 9, lines 385-387)
This manuscript is a resubmission of an earlier submission. The following is a list of the peer review reports and author responses from that submission.
Round 1
Reviewer 1 Report
Comments and Suggestions for Authors
General comments:
The manuscript entitled “Progressive exaptation of endogenous retroviruses in placental evolution in cattle” is a very intriguing and interesting manuscript that I enjoyed very much. The general critiques I have are the following: the authors are addressing a hypothesis in this manuscript, but most of it is stated as fact, which can be confusing for the reader to determine what parts are facts and what parts are speculation. It is only in the last sentence of the conclusion that we are reminded that there’s a hypothesis to the manuscript, and it is for the reader to go back and parse out what are facts and what are assertions. For better readability and understanding, I suggest that the authors specifically address their hypothesis (using the word “hypothesis”) in each section used to support the hypothesis.
Specific comments:
Lines 190-200: could the authors explain the significance of the differences between BNC, TNC, and MNC? While they obviously have different numbers of nuclei, do they behave differently? Are there differences in behavior between BNC and TNC?
Line 179: define synepitheliochorial
Lines 268-271: placentation depends on retrovirus genes that may have replaced endogenous predecessor genes during evolution- please describe more clearly the evidence and how the authors reached the conclusion that the retroviruses replaced endogenous predecessor genes.
It seems like the authors sometimes use “bovine” to refer specifically to cattle, and other times to refer to the entire Bovidae family, which gets confusing. Could the authors clarify when they are using bovine (specific to cattle) and bovine (entire Bovidae family including sheep- i.e. lines 283-286)
Line 289: could the authors introduce bERVE-A more thoroughly.
Lines 305-6 are unclear, could the authors clarify the difference between the two statements in this sentence “Although ERVs in bovine placentation have been studied extensively, the involvement of ERVs on placentation and its functioning is still unclear.”
Similarly, lines 311-312: it is unclear what mechanism the authors are referring to.
Conclusions: “immunosuppressive envs”- immunosuppression is mentioned once in the review paper on line 158, it seems inappropriate to include such a strong, declarative statement in the conclusions when it wasn’t touched upon in the manuscript.
Conclusions: the conclusion paragraph consists mostly of declarations. Could the authors clarify what the “hypothesis” is referencing on line 381? It is easy to get lost in the manuscript and forget it is addressing a hypothesis.
Comments on the Quality of English Language
Minor proofreading
Author Response
General comments:
The manuscript entitled “Progressive exaptation of endogenous retroviruses in placental evolution in cattle” is a very intriguing and interesting manuscript that I enjoyed very much. The general critiques I have are the following: the authors are addressing a hypothesis in this manuscript, but most of it is stated as fact, which can be confusing for the reader to determine what parts are facts and what parts are speculation. It is only in the last sentence of the conclusion that we are reminded that there’s a hypothesis to the manuscript, and it is for the reader to go back and parse out what are facts and what are assertions. For better readability and understanding, I suggest that the authors specifically address their hypothesis (using the word “hypothesis”) in each section used to support the hypothesis.
Thank you for taking your time to review our manuscript. As you pointed out, we have rewritten those to distinguish between facts and hypotheses throughout the manuscript.
Specific comments:
Lines 190-200: could the authors explain the significance of the differences between BNC, TNC, and MNC? While they obviously have different numbers of nuclei, do they behave differently? Are there differences in behavior between BNC and TNC?
To explain the differences between MNC, BNC and TNC, we have added the section of ‘Morphology of the Bovine Placenta’ (Revised manuscript, pages 5-6), and our response to the reviewer’s comment is as follows; In cattle and other ruminants, BNC begin to emerge around day 20 of gestation. They originate from trophoblast mononuclear cells (MNC) through processes such as mitotic polyploidization or endoreduplication [54, 57]. This process continues throughout gestation, primarily in cotyledons after day 30, resulting in a consistent 15 to 20% of the trophectoderm comprising BNC at various gestational stages [58]. In the placentome, which consists of maternal curancles and fetal cotyledons, BNC are believed to migrate while maintaining tight junctions with other trophoblast cells. Subsequently they fuse with a uterine luminal epithelium (LE) cell, leading to the formation of TNC in the sub-epithelial-stromal areas [54, 57]. This allows secretory granules containing unique placenta-specific hormones, such as placental lactogen and pregnancy-associated glycoproteins, as well as microvesicles, including exosomes, and other factors, to be re-leased via exocytosis into the endometrial stroma throughout gestation [54, 59].
Line 179: define synepitheliochorial
According to your comment, the definition of synepitheliochorial has been added in the section of Morphology of the Bovine Placenta (pages 5-6). The 'syn-' prefix signifies the contribution of BNC-derived syncytial trophoblast, while the term 'epitheliochorial' underscores the presence of extensive regions with simple placental cell attachment in the definitive placenta [55]. (Revised manuscript, page 6, lines 233-235)
Lines 268-271: placentation depends on retrovirus genes that may have replaced endogenous predecessor genes during evolution- please describe more clearly the evidence and how the authors reached the conclusion that the retroviruses replaced endogenous predecessor genes.
Although the following is not well discussed in the present manuscript, in primate evolutions, including humans, syncytin-2 entered the lineages approximately 40 MYA, followed by syncytin-1 integration 25 MYA (Imakawa et al., 2022). Similarly, in Bovidae, syncytin-Rum1 integrated into the bovine genome 30 MYA, followed by BERV-K1 25.4-18.3 MYA (Carter 2014). Syncytin-Rum1 exerts fusogenic activity only under acidic (pH 5.0) but not neutral (pH 7.0) conditions (Cornelis et al., 2013) whereas BERV-K1 exhibits much more fusogenic activity than Syncytin-Rum1 under physiological conditions (Nakaya et al., 2013). Similarly in the human, syncytin-1 has much more fusogenic activity than syncytin-2. Integration of Syncytin-Rum1, followed by BERV-K1 into the bovine genome, suggests that successive integrations of ERVs with homologous functions could occur at least in the bovine and humans. These observations suggest that newly acquired ERVs could function placental morphogenesis with greater efficacy than the preexisting genes. Moreover, these successive ERV acquisitions, therefore, are called a ‘baton pass’ hypothesis (Imakawa et al., 2015). Our response has been inserted in the revised manuscript, page 7, lines 302-308.
It seems like the authors sometimes use “bovine” to refer specifically to cattle, and other times to refer to the entire Bovidae family, which gets confusing. Could the authors clarify when they are using bovine (specific to cattle) and bovine (entire Bovidae family including sheep- i.e. lines 283-286)
Thank you for the explanation and we have clarified the terms in the revised manuscript.
Line 289: could the authors introduce bERVE-A more thoroughly.
While limited information was available regarding bERVE-A, we have added details about bERVE-A in the revised manuscript as follows; bERVE-A exhibited approximately 94% homology to a sequence similar to human syncytin and featured an ORF comprising 107 amino acids [48]. bERVE-A env was highly expressed in post-implantation conceptuses (D20 and D22), with no expression was observed in tissues or organs other than the ovary, testis, rectum, or spleen (Figure 2). Expression of bERVE-A was detected at a low level in the conceptuses from day 17 to 19 of gestation, with an increase in expression observed in fetal membranes up to day 30 of gestation [48]. bERVE-A is expressed preferentially in BNCs [56]. In addition, bERVE-A was rarely expressed in the endometrium throughout the estrus cycle [48]. This gene contains a syncytin 1-like SU domain and ASCT2 binding domain, but it does not pos-sess fusogenic activity due to the loss of the fusion peptide [48]. It is suggested that bERVE-A may have arisen as a result of BNC formation rather than serving as an initi-ator of cell-to-cell fusion, as it lacks an intact envelope sequence [78]. (Revised manuscript, page 8, lines 361-372)
Lines 305-6 are unclear, could the authors clarify the difference between the two statements in this sentence “Although ERVs in bovine placentation have been studied extensively, the involvement of ERVs on placentation and its functioning is still unclear.”
We have changed the sentence as follows; While the expression of ERVs in the bovine placenta has been extensively studied, their role in placentation and its functioning remains unclear. Furthermore, there has been limited analysis of the regulatory mechanisms governing ERV expression in bovine. This lack of clarity can be attributed, in part, to the scarcity of research on the cellular and molecular mechanisms of bovine placentation in comparison to other animal models. (Revised manuscript, page 8, lines 381-386)
Similarly, lines 311-312: it is unclear what mechanism the authors are referring to.
According to your comment, we have changed the sentence as follows; The study [79] demonstrated the mechanism associated with the differentiation of trophoblast MNC into BNC. (Revised manuscript, pages 8-9, lines 388-390)
Conclusions: “immunosuppressive envs”- immunosuppression is mentioned once in the review paper on line 158, it seems inappropriate to include such a strong, declarative statement in the conclusions when it wasn’t touched upon in the manuscript.
We agree with your comments and we have deleted the sentence related to immunosuppressive envs.
Conclusions: the conclusion paragraph consists mostly of declarations. Could the authors clarify what the “hypothesis” is referencing on line 381? It is easy to get lost in the manuscript and forget it is addressing a hypothesis.
As you pointed out, we have rewritten the Conclusions as follow; Throughout the course of evolution, organisms transitioned from asexual to sexual reproduction and subsequently shifted from oviparity to viviparity. At this time, an organ called the uterus was needed to nurture the fetus. This shift necessitated the emergence of the uterus as a vital organ for nurturing developing fetuses. During this crucial period, it is plausible that various retroviruses spread across mammalian species, integrating into their genomes. It is plausible that, prior to the emergence of the placenta, various retroviruses spread across mammalian species, integrating into their genomes. Subsequently, theses proviruses underwent both negative and positive selection, eventually acquiring functional roles in numerous cell types and organs. Most ERVs are not confined to specific tissues but instead exhibit varying degrees of expression across a wide range of tissues and organs. In essence, ERVs with pre-existing functions in di-verse tissues or organs might have been co-opted to operate at the fetal-maternal boundary within the uterus over the course of pregnancy evolution. This adaptation enabled new functions, such as cell fusion activities, to emerge over time, contributing to the remarkable diversity we observe in placental structures today. A comprehensive investigation of this hypothesis could shed light on the evolutionary history of the placenta and unravel the mysteries surrounding its diversity. (Revised manuscript, pages 10-11, lines 442-458)
Reviewer 2 Report
Comments and Suggestions for Authors
Abstract
Poorly written. Placenta was presumably present from the outset of eutherian mammals?
Does placentation increase survival compared to oviparity? What is the survival rate of monotreme embryos?
Presumably, pregnancy is more dangerous to the mother than laying eggs?
Many of the questions raised are poorly formulated, have already been addressed, etc.
Introduction
L45. Define ‘refined’.
L49-53. Assertions need to be supported by references.
Diversity and classification of placenta and subsequent sections:
L78. ‘retrotransposons’ or transposons, or both?
This section lacks depth and coherence; its attribution of ‘placenta’ to Mika et al., 2021 is misleading as the main focus of that paper is endometrium.
There are vague statements about the lack of explanations on why placentas are so diverse in phylogeny. However, in my opinion this mystery has already been solved and the solution is the rapid evolution and diversification arising from maternal-fetal conflict (Haig, 1993). The lack of any reference to (or rebuttal of) this theory makes me sceptical of the authors’ knowledge or the credibility of their judgement in this area.
L163-8. Hypothesis that ERVs explain differences in placental morphology is poorly expressed and supported.
The major subsequent sections are extensively descriptive, but the detail is overwhelming and no clear argument emerges. These sections could be summarized in Tables or Figures for greater clarity and concision. There are apparently new transcriptomics data which are undoubtedly worth reporting but the somewhat detached from the major part of the manuscript. Also, these data don’t support a specific role for most of the ERVs in placental development. If anything, they falsify the stated hypothesis.
The conclusions are rather vague, inconsequential and lack novelty. I think more condensed introductory sections and a greater focus on the novel data would be beneficial.
Comments on the Quality of English LanguageNo major issues.
Author Response
The authors sincerely appreciate your review and valuable suggestions.
Our responses to your specific comments are as follows:
Abstract
Poorly written. Placenta was presumably present from the outset of eutherian mammals?
Does placentation increase survival compared to oviparity? What is the survival rate of monotreme embryos?
Presumably, pregnancy is more dangerous to the mother than laying eggs?
Many of the questions raised are poorly formulated, have already been addressed, etc.
According to your comments, we have rewritten the Abstract, which was also read by a Native speaker.
It is our belief that placentas did not initially exist in eutherian mammals. This is because, as discussed in the manuscript, the uterus, not the oviduct, needed to evolve to form the placenta, and the expression of genes at the fetal-maternal boundary was also necessary (Revised manuscript, page 1, lines 39-42). The acquisition of placental-like structures is attributed to the acquisition of the paternally expressed gene 10 (Peg10) derived from a long terminal repeat- (LTR-) type retrotransposon approximately 160 million years ago (Revised manuscript, page 5, lines 189-191). The oldest eutherian, Juramaia sinensis, acquired a placenta around 160 million years ago, while the oldest known mammal today, Adelobasileus cromptoni, dates back to about 225 million years ago (Revised manuscript, page 2, lines 53-55).
We do not possess specific information on the survival rates of monotreme embryos, making it challenging to provide precise data. The survival rate of monotreme embryos can vary widely due to factors such as predation, environmental conditions, parental care, or habitat. Maternal care, including incubation and the protection of eggs and young, significantly enhances the chances of survival for monotreme embryos. Viviparity enables species to offer protection against environmental threats and facilitates the production of larger offspring with higher survival rates compared to oviparous species. Externally laid eggs are more vulnerable to threats than embryos in pouches or fetuses in uteri. Therefore, the survival probability of monotreme offspring may be lower compared to the offspring of therian mammals. Additionally, the maternal risk is lower in monotremes than in eutherian mammals.
It is also challenging to incorporate these revisions into the abstract due to word limit constraints. Furthermore, since this information diverges from the primary focus of the review, it is not included in the revised manuscript.
Introduction
L45. Define ‘refined’.
We have changed the sentences as follow; Theropod mammals, a lineage that encompassing marsupials and eutherians (all extent mammals except monotremes and marsupials), evolved the capacity for live birth (viviparity). During this evolutionary process, they lost both the mineralized eggshell and yolk were lost, and ultimately developing a placenta.
L49-53. Assertions need to be supported by references.
We have added the references as follows; The oldest eutherian, Juramaia sinensis, acquired a placenta approximately 160 million years ago [5], while the oldest mammal known today, Adelobasileus cromptoni, has been dated to about 225 million years ago [6]. (Revised manuscript, page 2, lines 53-55).
[5] Luo ZX, Yuan CX, Meng QJ, Ji Q. A Jurassic eutherian mammal and divergence of marsupials and placentals. Nature. 2011 476:442-445.
[6] Bi S, Zheng X, Wang X, Cignetti NE, Yang S, Wible JR. An Early Cretaceous eutherian and the placental-marsupial dichotomy. Nature. 2018 558:390-395.
Diversity and classification of placenta and subsequent sections:
L78. ‘retrotransposons’ or transposons, or both?
We think 'both' would be correct in this case. DNA transposons have not yet been characterized during early development. In most mammals, retrotransposons are the predominant TEs. In this review, we are focusing on the ERVs, which originated from LTR-retrotransposons. Therefore, we have purposely described it as 'retrotransposons'. In the revised manuscript, we have added a new section titled ‘The principle of genomic rewiring during waves of retroviral infection’.
This section lacks depth and coherence; its attribution of ‘placenta’ to Mika et al., 2021 is misleading as the main focus of that paper is endometrium.
As you pointed out, the paper by Mika and colleagues was about the endometrium, not the placenta. In the revised manuscript, we have changed placenta to endometrium or endometria as follows; They compared the genes used by the human endometrium with those expressed by endometria of 32 other species, including monkeys, marsupials, other mammals, birds, and reptiles [10]. (Revised manuscript, page 2, lines 94-96)
There are vague statements about the lack of explanations on why placentas are so diverse in phylogeny. However, in my opinion this mystery has already been solved and the solution is the rapid evolution and diversification arising from maternal-fetal conflict (Haig, 1993). The lack of any reference to (or rebuttal of) this theory makes me sceptical of the authors’ knowledge or the credibility of their judgement in this area.
Thank you very much for your valuable suggestion. We have added the sentences as follows; As pointed out by Haig [26], the phenotypic effects of a gene in one organism may influence the fate of the same gene in another organism. Therefore, the destiny of a gene holds greater evolutionary significance than that of the individual organism. While the emergence of changes in the expression of existing genes can be attributed to alterations in evolutionary interests across species and can be anticipated as a facet of placental functional evolution, this factor alone does not fully account for the diversity observed in placental morphology. Furthermore, it is plausible that maternal-fetal conflicts could lead to the removal of previously infected ERVs. Additionally, the acquisition of new genes, including ERVs, in a species-specific manner may contribute further to the generation of morphological diversity in the placenta (see Figure 1 in the revised manuscript). (Revised manuscript, pages 4-5, lines 175-184)
L163-8. Hypothesis that ERVs explain differences in placental morphology is poorly expressed and supported.
We have re-drawn the figure as Figure 1 and re-written the legend accordingly (page 3, lines 114-124).
The major subsequent sections are extensively descriptive, but the detail is overwhelming and no clear argument emerges. These sections could be summarized in Tables or Figures for greater clarity and concision. There are apparently new transcriptomics data which are undoubtedly worth reporting but the somewhat detached from the major part of the manuscript. Also, these data don’t support a specific role for most of the ERVs in placental development. If anything, they falsify the stated hypothesis.
According to your and the reviewer 4’s comments, we have added two Tables in the revised manuscript. One is the classification of mammalian placentas as Table 1 (page 3), the other is ERV genes supposed to play roles in placentation in the bovine as Table 2. (page 9)
The conclusions are rather vague, inconsequential and lack novelty. I think more condensed introductory sections and a greater focus on the novel data would be beneficial.
As you pointed out, we have evaluated the entire contents including Abstract and Conclusion, we have re-written the entire contents in the revised manuscript.
Reviewer 3 Report
Comments and Suggestions for Authors
This is an interesting and clearly written review that discusses evidence supporting the hypothesis that endogenous retroviruses (ERVs) have contributed to the evolution and diversity of different tissues and organs including those at the fetal-maternal boundary giving rise to placental diversity.
Author Response
Thank you for taking your time to review our manuscript.
Reviewer 4 Report
Comments and Suggestions for Authors
Sakurai et al. in their manuscript (Progressive exaptation of endogenous retroviruses in placental evolution in cattle) provided an overview of the current knowledge on fusogenic retroviral proteins in the genome of domestic cows and sheep. The authors highlighted the evolutionary aspects and mechanisms of progressive exaptation of newly acquired retroviral glycoproteins for new functions in placental development. Ruminants or sensu stricto bovines represent the best studied group in terms of identifying endogenous retroviruses expressed in the placenta and involved in placental functions, rivaled only by humans and mice. The richness of the involved bovine endogenous retroviruses (BERVs) allows hypotheses to be made about the gradual evolution and switching between different copies of BERVs. Thus, this topic deserves a review and the comparison of ruminants with other viviparous mammals is very intriguing. I suggest several points on how this review could be improved and made more illustrative for the reader.
Specific concerns
1. In comparing cattle to other mammals, several topics are missing from this review. The first step in the exaptation of endogenous retroviruses is the capture of fusogenic envelope glycoproteins. Next, new regulatory circuits are required to ensure placenta-specific expression and to limit expression of fusogenic proteins in nonplacental tissues. The authors do not describe which transcription factors regulate BERV promoters (except to mention the Wnt pathway). Nor do they analyze the epigenetic regulatory level.
2. In humans, negative regulation of fusogenic retroviral glycoproteins has been described for another endogenous retrovirus, suppressyn. In cattle, such BERV(s) has not yet been identified, however, this degree of exaptation in placental evolution should be considered and discussed.
3. The principle of genomic rewiring during waves of retroviral infection helps to understand their exaptation and the occurrence of evolutionary novelties. This mechanism should be discussed.
4. The chapter Expression of endogenous env-derived retroviruses in bovine placenta is too long and goes into unnecessary details. For example, on lines 244-280, the authors describe the identification of individual BERV copies and their expression without making any generalizations or conclusions. Also, in Figure 2, rather original data are presented that are not suitable for a review article. On the other hand, the review would have deserved more figures showing placental types, cell fusions in bovine placenta, and schematics of BERV proviral sequences.
Minor points
1. Several sentences (lines 32-35, 39-42, 295-296, 322-324 e.g.) are misleading, equivocal, or hard to understand. Please, clarify.
2. What do Peg10 and Peg11/Rtl1 encode? Please, specify.
3. Line 120: the word gene is duplicated.
4. The statement on lines 149-150 needs a reference.
5. Line 164: lineage instead of species would be better.
6. The statement on lines 234-237 needs a reference.
7. The reference 55 on line 239 is not correct because endogenous lentiviruses and deltaretroviruses were discovered much later.
8. The reference 56 on line 251 is not correct, There should be the reference 57.
9. Jagsekte retrovirus on line 281 should be Jagsiekte.
Comments on the Quality of English LanguageSeveral parts of the manuscript are hard to understands, some sentences should be rephrased to be clear and unequivocal. See minor points of the review.
Author Response
Sakurai et al. in their manuscript (Progressive exaptation of endogenous retroviruses in placental evolution in cattle) provided an overview of the current knowledge on fusogenic retroviral proteins in the genome of domestic cows and sheep. The authors highlighted the evolutionary aspects and mechanisms of progressive exaptation of newly acquired retroviral glycoproteins for new functions in placental development. Ruminants or sensu stricto bovines represent the best studied group in terms of identifying endogenous retroviruses expressed in the placenta and involved in placental functions, rivaled only by humans and mice. The richness of the involved bovine endogenous retroviruses (BERVs) allows hypotheses to be made about the gradual evolution and switching between different copies of BERVs. Thus, this topic deserves a review and the comparison of ruminants with other viviparous mammals is very intriguing. I suggest several points on how this review could be improved and made more illustrative for the reader.
Specific concerns
- In comparing cattle to other mammals, several topics are missing from this review. The first step in the exaptation of endogenous retroviruses is the capture of fusogenic envelope glycoproteins. Next, new regulatory circuits are required to ensure placenta-specific expression and to limit expression of fusogenic proteins in nonplacental tissues. The authors do not describe which transcription factors regulate BERV promoters (except to mention the Wnt pathway). Nor do they analyze the epigenetic regulatory level.
Unlike other species, there are no publications on the regulation of BERV expression, making it difficult to discuss. Kitao et al. (Retrovirology 2021) found functional syncytin post-transcriptional regulatory element (SPRE)-like elements in several syncytin genes derived from retroviruses and reported that these elements may promote expression of viral genes that had been repressed due to inefficient codon frequencies or repressive elements in the coding sequence. This suggests that SPRE-like elements are also present in syncytin-like genes such as BERV-K1, syncytin-rum1, and BERV-P in the bovine. It was also found that transcription factor binding site for Rxra was evaluated in the thyroid grand, however, it has not been evaluated in the uterus or placenta [Garcia-Etxebarria K and Jugo BM.]. For this reason, the finding was not used in the present manuscript.
- In humans, negative regulation of fusogenic retroviral glycoproteins has been described for another endogenous retrovirus, suppressyn. In cattle, such BERV(s) has not yet been identified, however, this degree of exaptation in placental evolution should be considered and discussed.
Suppressyn (SUPYN) is a truncated envelope protein derived from the HERV-H family of viruses (ERVH48-1) that is expressed at multiple sites in the human placenta, including the cytotrophoblast (CTB), and to a lesser extent the syncytiotrophoblast, of the floating villi, in the intermediate CTB of the anchoring villi, in invasive extravillous cytotrophoblast cells (EVT) within the decidua and in the endovascular trophoblast lining the maternal decidual vessels [Sugimoto J. 2013; 2019; Frank JA 2022]. SUPYN binds to the receptor for syncytin-1, neutral amino acid transporter, alanine serine cysteine transporter 2 (ASCT2, also known as SLC1A5), results in the inhibition of the CTB cell fusion [Sugimoto J. 2013; 2021]. It should be noted that SUPYN or SUPYN like sequence does not exist in the bovine genome. Thus, the relationship between SUPYN and BERV(s) cannot be discussed although a sequence(s) that counteracts with BERVs could exist.
- The principle of genomic rewiring during waves of retroviral infection helps to understand their exaptation and the occurrence of evolutionary novelties. This mechanism should be discussed.
Thank you very much for your comments. We agree with your comments and have added the section ‘The principle of genomic rewiring during waves of retroviral infection’ after the section of ‘Diversity and Classification of Placenta in Mammals’ as follows (page 4, lines 143-165).
Although different kinds of TEs are found in various organisms, all major TEs can be classified into two main categories, retrotransposons and DNA transposons [12]. Because TEs themselves are diverse and TEs make up a substantial portion of animal genomes, comprising approximately 27% in bovine genome [13]. TEs are pivotal in driving evolutionary changes in genome size and composition [14]. Host organisms share numerous repetitive sequences, encompassing genes, cis-regulatory elements, and chromatin domain boundaries. These shared sequences have the potential to alter gene regulatory networks and, moreover, are partially responsible for morphological evolution as in mammals [15]. Endogenous retroviruses (ERVs) belong to the TE and are considered to have originated from long terminal repeat- (LTR-) retrotransposons [16]. The genetic changes underlying the evolution of gestation appear to be associated with TEs that invaded and became integrated into early mammalian genomes. ERVs, in par-ticular, are thought to have originated from ancient viruses that infected primordial germ cells or their products, including oocytes, sperm, and fertilized eggs in mammals and other vertebrates [17]. ERVs have become integral components of host genome, resulting from waves of retroviral infection in the host's ancestors, insertion of retroviral sequences into germline DNA, and vertical transmission from one generation to the next [17]. Remarkably, retrovirus-derived sequences account for 8% of the human ge-nome [18], 10% of the mouse genome [19] and 18% of the bovine genome [20]. A study on the location and expression of bovine ERVs, as well as the characteristics of neigh-boring genes, has revealed that up to 1,610 bovine genes contain bovine ERVs (BERVs) inserted within their introns [21]. Most of these BERVs are oriented in the antisense direction, and several genes located in proximity to BERVs appear to be associated with viral response and chromatin assembly [21].
- The chapter Expression of endogenous env-derived retroviruses in bovine placenta is too long and goes into unnecessary details. For example, on lines 244-280, the authors describe the identification of individual BERV copies and their expression without making any generalizations or conclusions. Also, in Figure 2, rather original data are presented that are not suitable for a review article. On the other hand, the review would have deserved more figures showing placental types, cell fusions in bovine placenta, and schematics of BERV proviral sequences.
Following your and Reviewer 2's feedback, we have incorporated two tables into the revised manuscript. Table 1 presents the classification of mammalian placentas and can be found on page 3 of the revised manuscript. Additionally, Table 2 on page 9 lists ERV genes believed to play roles in placentation in ruminants. Our analysis of ERV expression in cattle revealed that it is not exclusive to the placenta, as illustrated in Figure 2. This suggests that ERVs might have played diverse roles in various organs or cell types, with some potentially repurposed to serve in placental structures and functions. To account for this possibility, we have included Figure 2 in our manuscript.
Minor points
- Several sentences (lines 32-35, 39-42, 295-296, 322-324 e.g.) are misleading, equivocal, or hard to understand. Please, clarify.
We have rechecked the manuscript, which was also evaluated by a Native speaker.
Lines 32-35 (now lines 32-37): Viviparity, which enables the maternal maintenance of the embryonic environment, is a reproductive strategy observed in various groups of vertebrates, including bony fishes, elasmobranchs, amphibians, reptiles, and mammals [1]. This primitive structure and function of the placenta, facilitating viviparity, was acquired relatively early in the course of vertebrate evolution, with a substantial fraction of vertebrate species developing this capability [2, 3].
Lines 39-42 (now lines 42-45): For example, monotremes, including the platypus and echidna, are oviparous mammals. They lay thin, low-mineral eggs that typically hatch about ten days after being laid. During a six-month period in the uterus, the developing young is nourished by maternal secretions through a simple yolk sac placenta [4].
Lines 295-296 (now lines 370-372): It is suggested that bERVE-A may have arisen as a result of BNC formation rather than serving as an initiator of cell-to-cell fusion, as it lacks an intact envelope sequence [78].
Lines 322-324 (now lines 399-402): BERV-K1, BERV-K2, bERVE-A, syncytin-Rum1, and BERV-P have all been reported to originate from env regions homologous to syncytin. However, ERVs from other regions, such as gag and pol, which may also play a role in ruminant placentation, have not been identified or characterized.
- What do Peg10 and Peg11/Rtl1 encode? Please, specify.
We have specified Peg10 and Peg11/Rtl1 in the revised manuscript as follows; Peg10 and Peg11/Rtl1 are imprinted genes thought to be derived from a Ty3/Gypsy LTR retrotransposon family encoding gag- and pol-like domains [28-30] (page 5, 197-198).
- Line 120: the word gene is duplicated.
We have deleted “gene”. (Revised manuscript, page 5, line 190)
- The statement on lines 149-150 needs a reference.
We added two references as follows (The statement is now page 4, line 169-171);
[23] De Parseval N and Heidmann T. Human endogenous retroviruses: From infectious elements to human genes. Cytogenet. Genome Res. 2005, 110, 318–332.
[24] Vargiu, L, Rodriguez-Tomé P, Sperber GO, Cadeddu M, Grandi N, Blikstad V, Tramontano E, Blomberg, J. Classification and characterization of human endogenous retroviruses; mosaic forms are common. Retrovirology 2016, 13, 7.
- Line 164: lineage instead of species would be better.
Thank you very much. As you pointed out, we have corrected it. (Revised manuscript, page 3, line 119)
- The statement on lines 234-237 needs a reference.
The reference has already listed up. The reference [49] is “Cornelis G., Heidmann O., Degrelle S.A., Vernochet C., Lavialle C., Letzelter C., Bernard-Stoecklin S., Hassanin A., Mulot B., Guillomot M., et al. Captured retroviral envelope syncytin gene associated with the unique placental structure of higher ruminants. Proc. Natl. Acad. Sci. USA. 2013;110:E828–E837.”
- The reference 55 on line 239 is not correct because endogenous lentiviruses and deltaretroviruses were discovered much later.
We have changed the references (The reference is now [70]). The reference is “Johnson WE. Endogenous Retroviruses in the Genomics Era. Annu. Rev. Virol. 2015, 2, 135–159.”
- The reference 56 on line 251 is not correct, There should be the reference 57.
Thanks for pointing out. References have been renumbered throughout the revised manuscript.
- Jagsekte retrovirus on line 281 should be Jagsiekte.
We have corrected it as “Jaagsiekte”. (Revised manuscript, page 8, line 352)
Round 2
Reviewer 2 Report
Comments and Suggestions for Authors
The authors have made extensive revisions to the manuscript which make it easier to read. However, in my opinion it still does not make a coherent argument supported by objectively analyzed data.
Reviewer 4 Report
Comments and Suggestions for Authors
In the revised manuscript, authors appropriately addressed all my minor concerns and specific point No. 3. On the other hand, the chapter Expression of env-derived Endogenous Retroviruses in Bovine Placenta is still too long and full of unnecessary details.
My specific points 1 and 2 were on topics that the review omitted. Although these aspects of BERV exaptation have not been studied, the authors should at least briefly discuss that epigenetic regulation of BERV and suppressyn analogues likely exist and await description. The situation for human placentally expressed HERVs (syncytins) should be presented, and the partial data on BERV (which, incidentally, the authors presented in their rebuttal letter) should be discussed on this background.
Comments on the Quality of English LanguageQuality of English language improved significantly and my concerns on clarity of several sentences were mostly addressed.